# tTARGIT AAVs mediate the sensitive and flexible manipulation of intersectional neuronal populations in mice

Paul V Sabatini[1], Jine Wang[1,2], Alan C Rupp[1], Alison H Affinati[1], Jonathan N Flak[3], Chien Li[4], David P Olson[5,6], Martin G Myers[1,6]*

[1]Department of Internal Medicine, University of Michigan, Ann Arbor, United States; [2]Chinese academy, College of Medical Science, China Three Gorges University, Yichang, China; [3]Indiana Biosciences Research Institute, Indianapolis, United States; [4]Novo Nordisk Research Center, Seattle, United States; [5]Department of Pediatrics, University of Michigan, Ann Arbor, United States; [6]Department of Molecular and Integrative Physiology, Ann Arbor, United States

**Abstract** While Cre-dependent viral systems permit the manipulation of many neuron types, some cell populations cannot be targeted by a single DNA recombinase. Although the combined use of Flp and Cre recombinases can overcome this limitation, insufficient recombinase activity can reduce the efficacy of existing Cre+Flp-dependent viral systems. We developed a sensitive dual recombinase-activated viral approach: tTA-driven Recombinase-Guided Intersectional Targeting (tTARGIT) adeno-associated viruses (AAVs). tTARGIT AAVs utilize a Flp-dependent tetracycline transactivator (tTA) 'Driver' AAV and a tetracycline response element-driven, Cre-dependent 'Payload' AAV to express the transgene of interest. We employed this system in $Slc17a6^{FlpO}$; $Lepr^{Cre}$ mice to manipulate LepRb neurons of the ventromedial hypothalamus (VMH; LepRb$^{VMH}$ neurons) while omitting neighboring LepRb populations. We defined the circuitry of LepRb$^{VMH}$ neurons and roles for these cells in the control of food intake and energy expenditure. Thus, the tTARGIT system mediates robust recombinase-sensitive transgene expression, permitting the precise manipulation of previously intractable neural populations.

*For correspondence:
mgmyers@umich.edu

## Introduction

The molecular heterogeneity of the nervous system requires a rich toolkit for precise study of distinct cell populations. Together with Cre recombinase-expressing mice, the available suite of Cre-dependent viral vectors permits the manipulation of genetically identified neural types. However, this approach neither permits the study of subpopulations of cells expressing a particular Cre allele (*Hodge et al., 2019*; *Mickelsen et al., 2019*) nor permits the study of Cre-expressing cells within a defined Central nervous system (CNS) site while excluding the Cre-expressing cells in closely apposed brain regions.

Restricting transgene expression to cells that express two marker genes can overcome this challenge (*Luan and White, 2007*). Early attempts at this approach utilized gene-specific, Cre-sensitive transgene-expressing alleles in combination with Cre alleles driven by distinct genes (*Chen et al., 2011*). Another solution involves expressing two Cre fragments across different transgenes with distinct promoters, reconstituting active Cre only in cells that express both pieces (*Hirrlinger et al., 2009*). These approaches generally involve substantial investments of time and resources, however, as they require generating and interbreeding multiple new gene- and experiment-specific alleles.

Transgene expression can also be directed to cell types defined by two marker genes through the use of recombinase-dependent alleles (often inserted into the *Rosa26* Locus) containing a

**eLife digest** The brain contains hundreds of types of neurons, which differ in size, shape and behavior. But neuroscientists often wish to study individual neuronal types in isolation. They are able to do this with the aid of a toolkit made up of two parts: viral vectors and genetically modified mice.

Viral vectors are viruses that have been modified so that they are no longer harmful and can instead be used to introduce genetic material into cells on demand. To create a viral vector, the virus' own genetic material is replaced with a 'cargo' gene, such as the gene for a fluorescent protein. The virus is then introduced into a new host such as a mouse. Importantly, the virus only produces the protein encoded by its 'cargo' gene if it is inside a cell that also contains one of two specific enzymes. These enzymes are called Cre and Flp.

This is where the second part of the toolkit comes in. Mice can be genetically engineered to produce either Cre or Flp exclusively in specific cell types. By introducing a viral vector into mice that produce either Cre or Flp only in one particular type of neuron, researchers can limit the activity of the cargo gene to that neuronal type.

But sometimes even this approach is not selective enough. Researchers may wish to limit the activity of the cargo gene to a subpopulation of cells that produce Cre or Flp. Or they may wish to target only Cre- or Flp-producing cells in a small area of the brain, while leaving cells in neighboring areas unaffected.

Sabatini et al. have now overcome this limitation by developing and testing a new set of viral vectors that are active only in neurons that produce both Cre and Flp. The vectors are called tTARGIT AAVs and allow researchers to target cells more precisely than was possible with the previous version of the toolkit.

Sabatini et al. show tTARGIT AAVs in action by using them to identify a group of neurons that control how much energy mice use and how much food they eat. As well as applying the vectors to their own research on obesity, Sabatini et al. have also made them freely available for other researchers to use in their own projects.

---

ubiquitous promoter and tandem STOP cassettes that are each excised by distinct recombinases (*Daigle et al., 2018*; *Sciolino et al., 2016*).

Neuroscience research often requires a higher degree of spatial specificity than afforded by the intersectional genetic models outlined above. Stereotaxic injections of recombinase-sensitive viral vectors can restrict transgene expression to a narrow anatomical region. Adeno-associated viruses (AAVs) generally represent the preferred viral system for Cre-dependent transgene expression, given their minimal toxicity and the speed and ease of their generation. Developing AAVs sensitive to multiple recombinases has been challenging because of the limited AAV genome size, which precludes the use of multiple recombinase-sensitive STOP cassettes (*Wu et al., 2010*).

The INTRSECT system overcomes this limitation by utilizing a single AAV vector that flanks the transgene coding sequence with lox and FRT sites in such a way that combinatorial expression of Cre and Flp permits expression of a functional pre-mRNA that can then be spliced to produce a mature coding sequence (*Fenno et al., 2020*; *Fenno et al., 2014*). The multiple inversion and splicing steps involved in this system can limit transgene expression, however, perhaps due to the relatively poor recombinase activity of Flp (and even optimized FlpO). Furthermore, generating INTRSECT viruses that express new transgenes requires a relatively complex and labor-intensive design and optimization process (*Fenno et al., 2020*; *Fenno et al., 2017*).

Seeking a dual recombinase-activated AAV system to overcome these limitations and that could be modified quickly and easily, we generated tetracycline transactivator (tTA)-driven Recombinase-Guided Intersectional Targeting (tTARGIT) AAVs composed of a Flp-dependent tetracycline transactivator (tTA) 'Driver' AAV and a tetracycline response element (TRE)-driven Cre-dependent 'Payload' AAV to express the transgene of interest.

We applied tTARGIT AAVs to the study of ventromedial hypothalamic LepRb (VMH; LepRb$^{VMH}$) neurons, which modulate metabolic adaptations to obesogenic diets (*Bingham et al., 2008*; *Dhillon et al., 2006*) but have proven difficult to study directly due to the density and proximity of neighboring LepRb populations. Together with *Lepr$^{Cre}$* and *Slc16a7$^{FlpO}$*, tTARGIT AAVs allowed us

to overcome these challenges and reveal a specific role for LepRb$^{VMH}$ neurons in suppressing food intake and increasing energy expenditure to promote weight loss.

## Results

The density of LepRb neurons in the adjacent arcuate nucleus (ARC), dorsomedial hypothalamus, and lateral hypothalamic area complicated our initial attempts to study LepRb$^{VMH}$ neurons using *Lepr$^{Cre}$* mice and Cre-dependent vectors; viruses targeted to the VMH spread to other nearby *Lepr-$^{Cre}$* neurons, confounding the interpretation of our results (data not shown). Because *Slc17a6*, encoding the vesicular glutamate transporter 2 (vGLUT2) protein, expression is largely restricted to LepRb$^{VMH}$ neurons and absent from most surrounding LepRb cells (*Vong et al., 2011*), we generated a *Slc17a6$^{FlpO}$* strain and crossed it with *Lepr$^{Cre}$* and a novel Flp- and Cre-dependent reporter (*Rosa26$^{RCFL-eGFP-L10a}$*). We then tested the potential ability of this combination of Cre and Flp alleles to specify LepRb$^{VMH}$ neurons in the mediobasal hypothalamus. Although these reporter mice identified Cre- and Flp-co-expressing LepRb$^{Slc17a6}$ neurons elsewhere in the brain, within the mediobasal hypothalamus this approach largely limited eGFP expression to VMH cells (*Figure 1—figure supplement 1*).

We thus sought to use Flp- and Cre-dependent AAVs to target LepRb$^{VMH}$ neurons and omit manipulation of non-VMH LepRb$^{Slc17a6}$ cells. We injected the INTRSECT AAV system (*Fenno et al., 2014*) into the VMH of *Slc17a6$^{FlpO}$;Lepr$^{Cre}$* mice in an attempt to express channelrhodopsin (ChR2) in LepRb$^{VMH}$ cells. However, this approach resulted in detectable ChR2 expression in one or fewer cells per section (*Figure 1—figure supplement 2*). We surmised that while INTRSECT works well in systems with robust Flp and Cre activities, the poor recombinase activity of Flp and the more moderate Flp and Cre expression mediated by *Slc17a6$^{FlpO}$* and *Lepr$^{Cre}$*, respectively, might limit INTRSECT-mediated transgene expression in LepRb$^{VMH}$ cells.

We therefore set out to develop a more sensitive AAV system to drive Cre+Flp-dependent transgene expression, using as our framework a previously described inducible gene expression system based on recombinase-dependent expression of the tetracycline transactivator (tTA) in combination with a TRE-driven transgene-expressing allele (*Chan et al., 2017*; *He et al., 2016*). We packaged this system into two viral vectors, hereafter 'tTA-driven Recombinase-Guided Intersectional Targeting' (tTARGIT) AAVs.

Our tTARGIT system utilizes 'Driver' and 'Payload' AAVs. The Driver (AAV-hSYN1-fDIO-tTA) utilizes Flp-dependent Double-Floxed Inverted Open reading frame (fDIO) cassette (*Fenno et al., 2014*) to Flp-dependently invert tTA, permitting its expression by a human synapsin I (hSYN1) promoter. This virus also contains two tetracycline operators (TetO) to drive a positive feedback loop and increase tTA expression (*Chan et al., 2017*). The Payload AAV mediates tTA/TRE-dependent transgene expression following its Cre-mediated inversion into the sense orientation. Hence, only cells that contain both recombinases express the transgene (*Figure 1a*). Tetracycline inhibits tTA-dependent gene expression (*Das et al., 2016*), so this system mediates constitutive payload expression in target cells in the absence of tetracycline.

To test the recombinase dependence of this system, we combined the Flp-dependent Driver AAV and a Payload AAV that permits the tTA/TRE-mediated Cre-dependent expression of a ChR2-TdTomato fusion protein (ChR2-TdT; AAV-TRE-DIO-ChR2-TdT). We co-injected these viruses into the VMH of wild-type mice, mice that expressed either *Slc17a6$^{FlpO}$* or *Lepr$^{Cre}$* only, or *Slc17a6$^{FlpO}$; Lepr$^{Cre}$* mice (*Figure 1b,c*). We detected no TdT (DSRed-immunoreactivity [IR]) in the VMH of wild-type or *Slc17a6$^{FlpO}$* animals and minimal expression in *Lepr$^{Cre}$* mice (*Figure 1—figure supplement 3*). In contrast, the Driver+Payload combination mediated robust DSRed-IR in the VMH of *Slc17a6$^{FlpO}$;Lepr$^{Cre}$* mice (*Figure 1c*); furthermore, VMH photostimulation in these mice promoted robust colocalization of DSRed- and FOS-IR, consistent with the ability of this system to activate transduced Flp+Cre-expressing cells (*Figure 1—figure supplement 4*).

We also tested the requirement for both the Driver and Payload viruses in this system by injecting each virus alone, or both viruses together, into the VMH of *Slc17a6$^{FlpO}$;Lepr$^{Cre}$* animals (*Figure 1d*). As expected, injecting either virus alone produced minimal or no detectable DSRed-IR, while co-injection of the two viruses yielded robust DSRed-IR (*Figure 1e*). Thus, robust transgene expression by the tTARGIT system requires injection of both the Driver and Payload AAVs, as well as the presence of both Flp and Cre recombinases.

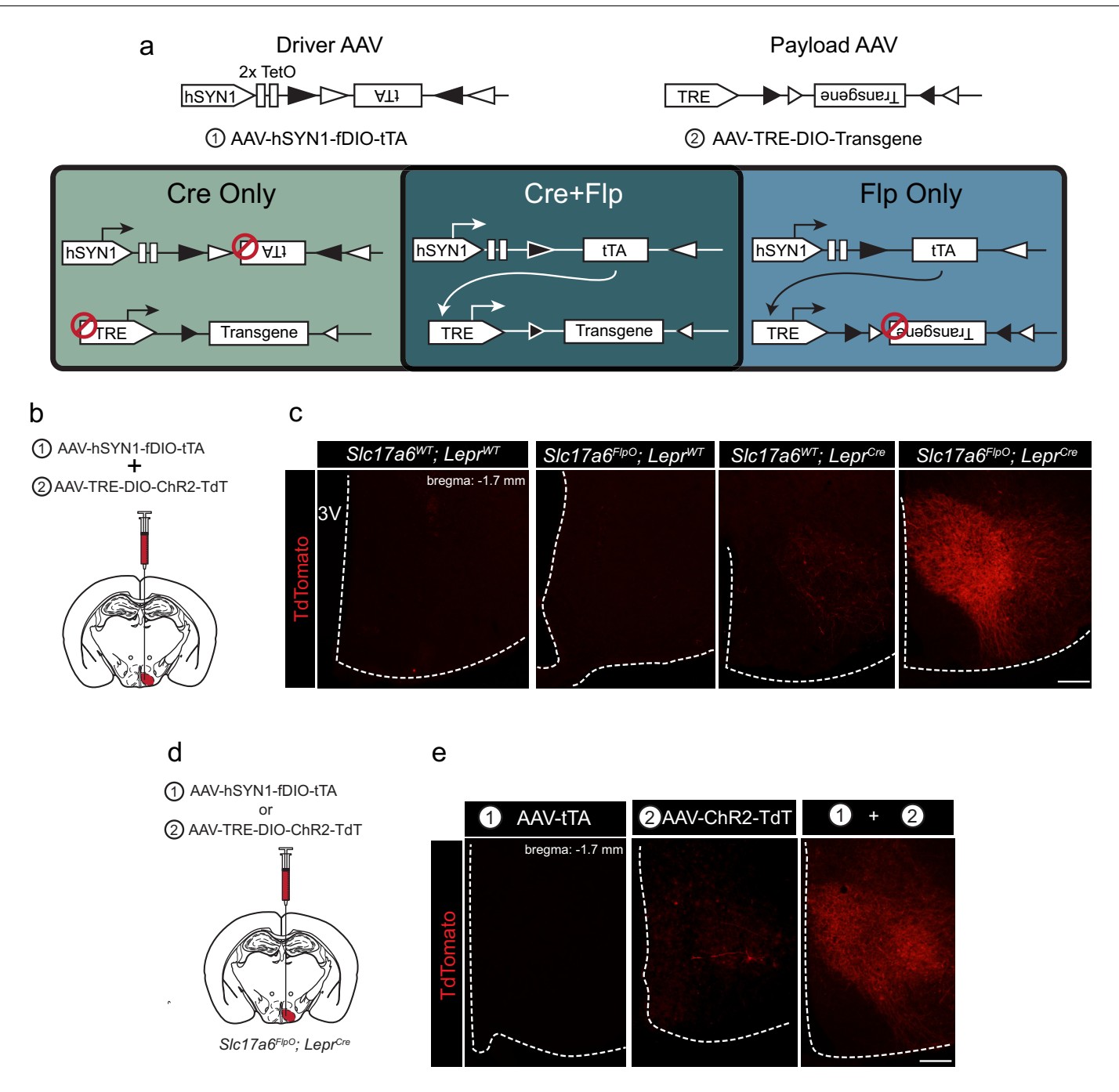

**Figure 1.** tTARGIT AAVs: a dual virus system to target intersectional populations. (**a**) The tTARGIT system employs the combination of 'Driver' (AAV-hSYN-fDIO-tTA) and 'Payload' (AAV-TRE-DIO-Payload) AAVs. The Driver virus encodes a Flp-dependent tetracycline transactivator (tTA) under control of the human synapsin I (hSYN1) promoter and two tetracycline operators (TetO). The Payload virus encodes a Cre-dependent Payload transgene under control of the tetracycline response element (TRE). (**b,c**) Experimental scheme (**b**) and representative images (**c**) showing the detection of TdTomato (DSRed-IR, red) following the co-injection of AAV-hSYN-fDIO-tTA and AAV-TRE-DIO-ChR2-TdT into the VMH of (from left to right, as labeled) wild-type (WT), *Slc17a6*$^{FlpO}$, *Lepr*$^{Cre}$, or *Slc17a6*$^{FlpO}$;*Lepr*$^{Cre}$ mice. (**d,e**) Experimental schematic (**d**) and representative images (**e**) showing the detection of TdTomato (DSRed-IR, red) following the injection of (from left to right, as labeled): (1) AAV-hSYN-fDIO-tTA, (2) AAV-TRE-DIO-ChR2-TdT, or the two viruses combined (1+2) into the VMH of *Slc17a6*$^{FlpO}$;*Lepr*$^{Cre}$ mice. Scale bars = 100 μm.

The online version of this article includes the following figure supplement(s) for figure 1:

**Figure supplement 1.** An intersectional approach for targeting LepRb$^{VMH}$ neurons.

**Figure supplement 2.** INTRSECT transgene expression in *Slc17a6*$^{FlpO}$;*Lepr*$^{Cre}$-defined LepRb$^{VMH}$ neurons.

*Figure 1 continued on next page*

**Figure supplement 3.** VMH hM3Dq-mCherry expression in *Slc17a6^FlpO;Lepr^Cre* and *Lepr^Cre*-only mice.

**Figure supplement 4.** FOS response to photostimulation of tTARGIT AAV-driven ChR2 production in LepRb^VMH cells.

**Figure supplement 5.** Design and validation of the tTARGIT AAV approach to Flp-ON/Cre-OFF genetics.

The tTARGIT approach can also be modified from a Flp-ON/Cre-ON system, requiring both Flp and Cre recombinases for payload expression, to a Flp-ON/Cre-OFF system, mediating transgene expression in all Flp-expressing cells that do not contain Cre (*Figure 1—figure supplement 5*; *Table 1*). Placing the payload transgene in the forward orientation within the DIO cassette permits tTA-driven transgene expression in all Flp-expressing cells that do not contain Cre because Cre inverts the Payload transgene into the antisense orientation. We tested this system with a novel Cre-inactivated Payload virus expressing a hM3Dq designer receptor exclusively activated by designer drugs (DREADD)-mCherry transgene. We co-injected this Cre-OFF Payload and the Flp-dependent Driver AAV into the VMH of *Slc17a6^FlpO;Lepr^Cre* animals on the Cre-dependent *Rosa26^LSL-GFP-L10a* background (*Figure 1—figure supplement 5b*). As expected, this modified tTARGIT system drove hM3Dq-mCherry expression almost exclusively in cells that did not express the Cre-dependent GFP (*Figure 1—figure supplement 5c*). We surmise the few GFP-IR neurons with detectable mCherry, constituting 1–7% of all mCherry cells, might result from the low Cre expression mediated by *Lepr^Cre* (*Patterson et al., 2011*) and predict that the Flp-On, Cre-Off tTARGIT system should demonstrate complete Payload inactivation when used in conjunction with a more robustly expressing Cre allele or require longer incubation periods to fully inactivate the payload transgene.

To define the projection targets of LepRb^VMH neurons, we developed a payload virus (AAV-TRE-DIO-GFP-2A-SynmRuby) that encodes GFP plus a cotranslationally expressed synaptophysin-mRuby transgene (*Figure 2a*). Co-injection of this tTARGIT Payload AAV and the Driver virus into the VMH of *Slc17a6^FlpO;Lepr^Cre* mice promoted robust VMH-restricted GFP-IR (*Figure 2b*). mRuby detection

**Table 1.** Available tTARGIT Vectors.

| Driver viruses | Notes |
| --- | --- |
| AAV-hSYN1-fDIO-tTA | Drives expression in the absence of DOX |
| AAV-hSYN1-fDIO-rtTA | Drives expression in the presence of DOX; not tested |
| Payload viruses | |
| Flp-ON/Cre-ON (Payload inverted) | |
| AAV-TRE-DIO-hM3Dq-mCherry | Tested |
| AAV-TRE-DIO-ChR2-TdTomato | Tested |
| AAV-TRE-DIO-Cas9 | Not tested |
| AAV-TRE-DIO-TVA-oG-mCherry | Not tested |
| AAV-TRE-DIO-eGFP-2A-TetanusToxin | Not tested |
| AAV-TRE-DIO-SwiChR-eYFP | Not tested |
| AAV-TRE-DIO-eGFP-L10a | Not tested |
| AAV-TRE-DIO-GCaMP6s | Not tested |
| AAV-TRE-DIO- proCaspase-3-TEVp | Not tested |
| AAV-TRE-DIO-GFP-2A-SynmRuby | Tested; GFP expression permits tracing; limited SynmRuby expression |
| Flp ON/Cre-OFF (Payload Sense orientation) | |
| AAV-TRE-DIO-hM3Dq-mCherry | Tested; may require strong Cre driver for complete inactivation in Cre neurons |
| AAV-TRE-DIO-ChR2-TdTomato | |
| AAV-TRE-DIO-eGFP-2A-TetanusToxin | |

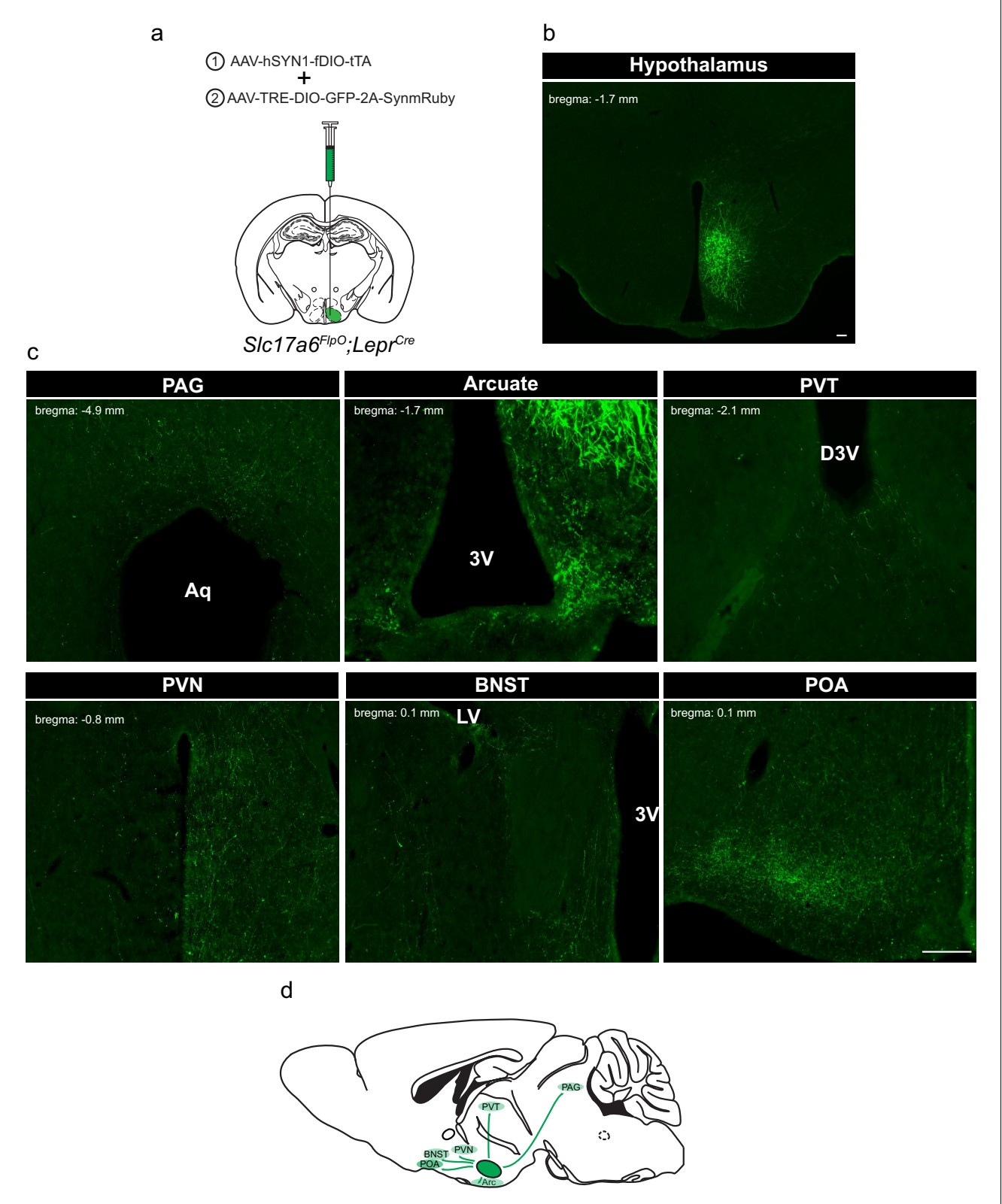

**Figure 2.** Defining the downstream projections of LepRb[VMH] neurons. (**a**) Experimental schematic showing the injection of the Driver AAV with the Payload AAV encoding a GFP-2A-SynmRuby transgene into the VMH of *Slc17a6[FlpO];Lepr[Cre]* animals. (**b,c**) Representative image of GFP-IR (green) showing viral transduction in the VMH (**b**) and projections (**c**) in the periaqueductal gray (PAG), arcuate nucleus (ARC), paraventricular hypothalamic nucleus (PVN), bed nucleus of the stria terminalis (BNST), and preoptic area (POA). (**d**) Cartoon showing the projection targets of LepRb[VMH] neurons.

(DSRed-IR) for this virus was much lower than for GFP, however (data not shown); thus, we used GFP-IR to detect projections from LepRb$^{VMH}$ cells. Assessing the entire CNS for the presence of GFP-IR revealed terminals in the periaqueductal gray, the arcuate, the periventricular thalamic nucleus, the periventricular hypothalamic nucleus, the bed nucleus of the stria terminalis, and the preoptic area (*Figure 2c,d*). These are consistent with the known projections of the VMH (*Canteras et al., 1994*; *Meek et al., 2016*; *Zhang et al., 2020*).

Ablating *Lepr* expression from the VMH (*Nr5a1$^{Cre}$*-mediated) promotes obesity associated with decreased energy expenditure in high-fat-diet-fed animals, suggesting a specific role for LepRb signaling in LepRb$^{VMH}$ cells in the control of energy balance via the dietary modulation of energy utilization (*Bingham et al., 2008*; *Dhillon et al., 2006*). To define the function of LepRb$^{VMH}$ cells, rather than the function of LepRb in these cells, we developed a Payload virus containing an inverted hM3Dq-mCherry transgene. We co-injected the Driver AAV and this AAV-TRE-DIO-hM3Dq-mCherry Payload virus into the VMH of *Slc17a6$^{FlpO}$;Lepr$^{Cre}$* animals (LepRb$^{VMH-Dq}$ mice; *Figure 3a*). This approach promoted robust VMH-restricted expression of functional hM3Dq-mCherry as administration of the DREADD activator (*Pei et al., 2008*), clozapine-N-oxide (CNO), stimulated colocalization of FOS with DSRed-IR cells (*Figure 3b–d*).

To determine the potential modulation of energy expenditure, activity, and food intake by LepRb$^{VMH}$ neuron activation, we placed LepRb$^{VMH-Dq}$ animals in metabolic cages and administered either vehicle or CNO twice daily (*Figure 3e–j*). Compared to vehicle administration, activating LepRb$^{VMH}$ neurons significantly increased 24 hr oxygen consumption (VO$_2$) and energy expenditure, both primarily due to effects during the light phase, despite decreasing ambulatory activity over 24 hr (primarily due to effects during the dark phase) (*Figure 3g,h*). Additionally, the hM3Dq-mediated activation of LepRb$^{VMH}$ neurons also suppressed 24 hr food intake, primarily due to decreased light-phase feeding, revealing a previously unsuspected role for these cells in the suppression of feeding. CNO also decreased the respiratory exchange ratio during the light phase, consistent with the increased metabolism of fat stores due to the combination of increased energy expenditure and decreased food intake.

To understand whether effects on brown adipose tissue (BAT) might contribute to the increased energy expenditure during LepRb$^{VMH}$ neuron activation, we placed temperature probes in the interscapular space of LepRb$^{VMH-Dq}$ animals to monitor BAT thermogenesis. Compared to controls, CNO significantly increased intrascapular temperatures in LepRb$^{VMH-Dq}$ animals, suggesting LepRb$^{VMH}$ neurons promote energy expenditure at least in part by augmenting BAT thermogenesis (*Figure 3k–l*).

As activating LepRb$^{VMH}$ neurons increased energy expenditure and decreased food intake, we surmised that these neurons should promote weight loss. We thus administered CNO in drinking water to LepRb$^{VMH-Dq}$ mice or *Lepr$^{Cre}$*-only control animals (lacking any Flp expression) injected with the tTARGIT hM3Dq for 3 days. During this time, the body weight of LepRb$^{VMH-Dq}$ mice decreased by approximately 10% (*Figure 4a*), returning to baseline following the cessation of CNO exposure. Importantly, body weight of *Lepr$^{Cre}$*-only controls remained stable. While CNO treatment decreased food consumption (largely during the second day of treatment) and water intake (*Figure 4b–d*),the magnitude and timing of these ingestive effects dictates that neither could account for the decreased body weight mediated by CNO, consistent with the notion that increased energy expenditure mediates the major effect of LepRb$^{VMH}$ cells on body weight.

Hence, the use of our dual recombinase-dependent tTARGIT AAV system permitted us to determine that LepRb$^{VMH}$ neuron activation increased energy expenditure and decreased food intake during the inactive phase, suggesting the diurnal control of energy balance by LepRb$^{VMH}$ neurons.

## Discussion

The use of sequence-specific DNA recombinases (Cre, Flp, and others) in conjunction with recombinase-dependent genetic alleles and viral vectors has revolutionized our ability to manipulate specific circuits and understand the central nervous system. The lack of robust viral systems to manipulate cell populations defined by the expression of multiple genes has impeded the study of more refined neural populations, however, including those identified by single cell RNA-sequencing (*Campbell et al., 2017*). Our tTARGIT AAV system addresses many shortcomings of previous

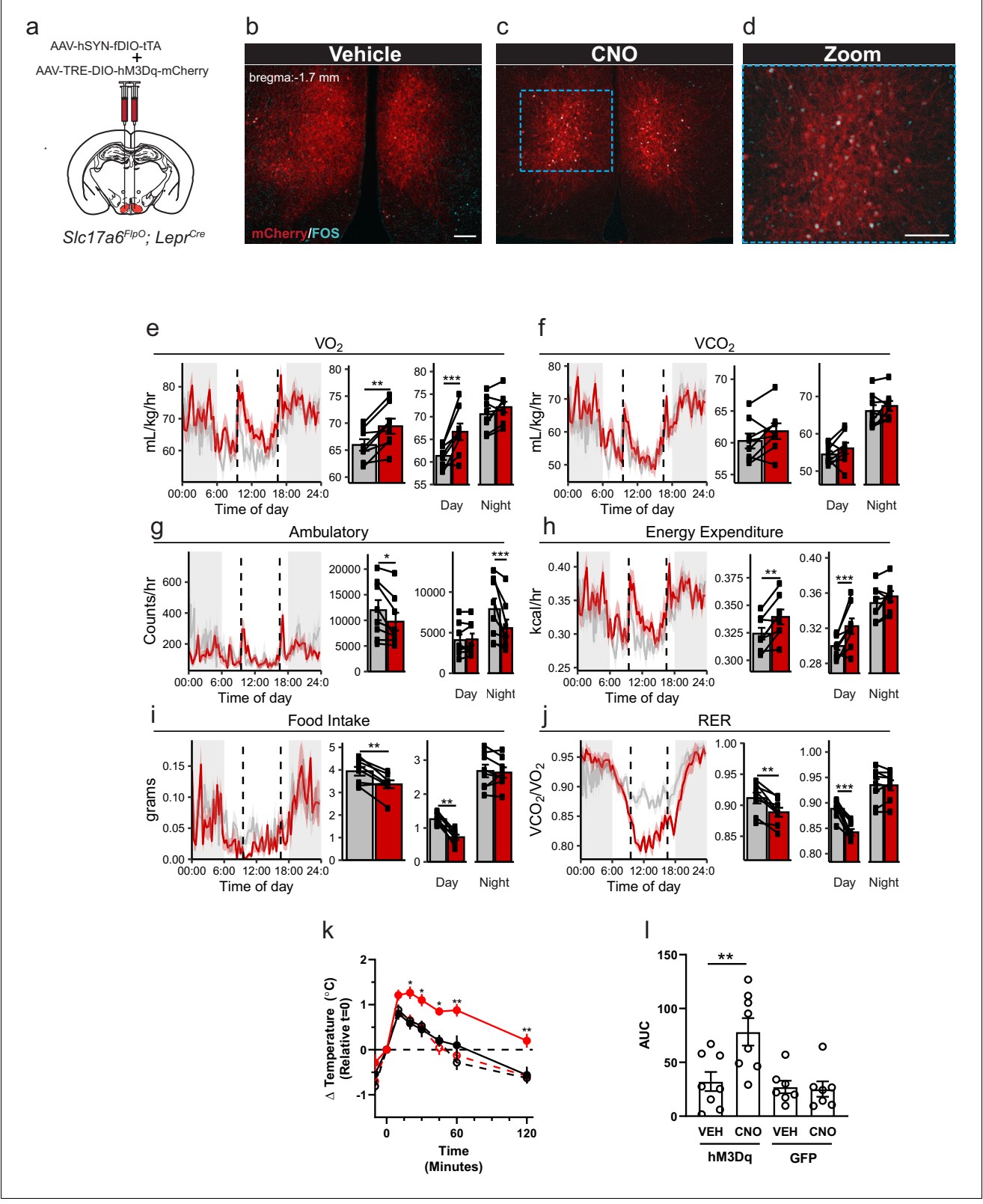

**Figure 3.** Activation of LepRb[VMH] neurons decreases food intake in addition to increasing brown adipose tissue (BAT) thermogenesis and energy expenditure. (**a**) Approach for activating LepRb[VMH] neurons by co-injecting the AAV-hSYN-fDIO-tTA Driver and the AAV-TRE-DIO-hM3Dq-mCherry Payload viruses into the VMH of *Slc17a6^FlpO;Lepr^Cre* animals. (**b–d**) Representative images showing mCherry detection (DSRed-IR, red) and FOS-IR (cyan) in LepRb[VMH-Dq] mice administered Vehicle (**b**) or CNO (1 mg/kg) (**c,d**). The right-hand panel of (**c**) shows a digital zoom of the boxed region in

*Figure 3 continued on next page*

*Figure 3 continued*

the left panel. Scale bars = 100 µm. (e–j) Results from metabolic cage analysis of LepRb$^{VMH-Dq}$ mice (n = 6) treated with either vehicle (gray) or 1 mg/kg CNO (red) at 9:30 and 16:30 (dotted lines). Lines in left panels denotes mean value; shading denotes SEM. Each animal was treated with vehicle for 2 days followed by CNO for 2 days to allow pairing. Bar graphs to the right show the average for each mouse at each time point across 24 hr and separated by time in light cycle (day = light, night = dark). (k, l) Changes to intrascapular temperatures over 120 min in LepRb$^{VMH-Dq}$ mice (n = 8) or GFP-injected (n = 7) controls following vehicle (gray) or CNO (red; 1 mg/kg) administration at 30˚C. (l) Shows area under the curve (AUC) for each treatment condition in (k). For metabolic cage studies, statistical significance was determined using either a paired t-test (full-day data) or a linear mixed model for effects by time of day. For interscapular temperature measurements, significance was determined by paired t-test. *p<0.05, **p<0.01, ***p<0.001.

The online version of this article includes the following source data for figure 3:

**Source data 1.** Raw Data from metabolic cages.

intersectional tools, including limitations to transgene expression and the difficulty of incorporating novel transgenes into the AAV plasmids.

As the tTARGIT system is based on the use of dual AAVs, it is possible that the two viruses could compete with each other for the surface receptors required for viral internalization and/or for the cellular machinery required for transgene production, thereby reducing expression efficiency from both AAVs. Based upon our results, and since dual AAVs have been used successfully previously (*Akil et al., 2019*; *Xu et al., 2018*; *Yang et al., 2016*) and transduce the same cells in vivo (*Vardy et al., 2015*), we surmise that if such competition exists with tTARGIT AAVs, it is minor.

To further facilitate the study of intersectional neural populations, we developed a suite of Cre-dependent tTARGIT Payload plasmids (*Table 1*). While we have not yet tested the function of all of these, our experience with the transgenes that we have tested predicts similarly robust expression of the various Payload AAV transgenes. The limitations of these Payload vectors are likely to mirror those of standard viruses with Cre-dependent transgenes, including the requirement for stoichiometric transduction of/recombination in the cell type of interest to observe the effects of interfering with neuronal function.

Additionally, the tTARGIT system displays some detectable payload expression, albeit at low levels and in a small number of cells, in *Lepr^Cre*-only mice co-injected with Driver and Payload AAVs, as well as in *Slc17a6^FlpO*;*Lepr^Cre* mice injected with the Payload AAV alone. We surmise this results from the previously noted tTA-independent activity of the TRE (*Das et al., 2016*). We expect this tTA-independent Payload transgene expression to predominately occur in Cre-expressing cells, which have inverted the transgene into the correct orientation downstream of the TRE. In our studies, leaky payload expression had little impact as the expression levels were low and failed to produce a metabolic response to CNO in drinking water. Nevertheless, the possibility of tTA-independent payload expression represents an important consideration for experimental design and requires the inclusion of appropriate controls.

In addition to Cre-ON/Flp-ON tTARGIT plasmids, we also generated Cre-inactivated Payload vectors to specifically mark Cre-negative Flp-expressing cells within the injection field. Within our model, we found up to 7% of mCherry-IR cells were co-labeled with the Cre-dependent GFP reporter. We posit that increasing post-injection incubation times prior to study or using alleles with higher Cre expression than observed in *Lepr^Cre* may further reduce the number of Cre-labeled cells expressing the Cre-OFF tTARGIT payload. Hence, inclusion of appropriate control will be required for studies employing tTARGIT AAVs.

While the tTARGIT AAV system as we have built it is designed to constitutively express transgenes in Cre- and Flp-expressing cells without the use of tetracycline-like compounds (usually doxycycline), it should be possible to decrease transgene expression from the tTARGIT AAVs by doxycycline treatment. Indeed, we built a Flp-dependent rtTA Driver virus that is predicted to require doxycycline treatment to mediate strong transgene expression. The rtTA-based system can drive low-level transcription independent of doxycycline (*Zhu et al., 2001*), however, and we have not yet tested this system.

The use of tTARGIT AAVs permitted us to target robust transgene expression to LepRb$^{VMH}$ neurons specifically, in isolation from LepRb neurons in adjacent hypothalamic nuclei. While deleting *Lepr* from the VMH of chow-fed mice or restoring VMH *Lepr* expression on an otherwise LepRb-deficient background minimally (if at all) alters energy balance, knockout mice fail to increase energy

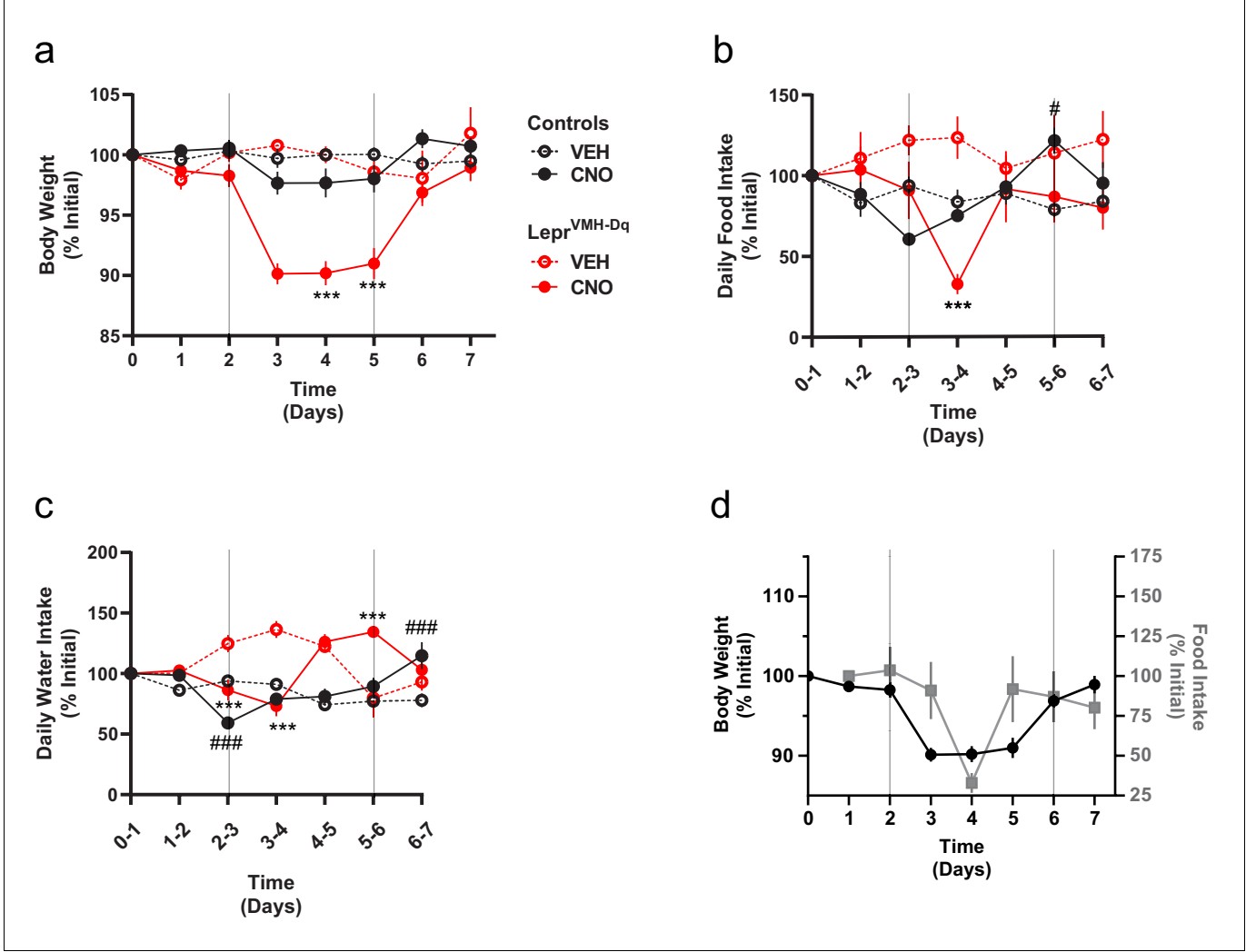

**Figure 4.** Chronically activating LepRb[VMH] neurons promotes negative energy balance. (a–c) Body weight, food intake, and water intake of chow-fed *Lepr[Cre]*-only controls injected with AAV-hSYN1-fDIO-tTA and AAV-TRE-DIO-hM3Dq-mCherry (black lines; VEH n = 5; CNO n = 8) and LepRb[VMH-Dq] mice (red lines n = 8 both conditions) receiving 2 days normal drinking water (days 0–2), followed by 3 days exposure to CNO-laced drinking water (days 2–5), followed by an additional 2 days of normal drinking water. Onset and termination of CNO treatment are denoted by vertical lines in each panel. Vehicle controls (dashed red and black lines) did not receive CNO on days 2–5. (d) Comparisons between body weight (closed black circles) and food intake (gray squares) over time in LepRb[VMH-Dq] animals. Data is presented as mean ± SEM. Significance was determined by linear mixed model; ***p<0.001 between CNO and VEH LepRb[VMH-Dq] animals. #p<0.05, ###p<0.001 between CNO and VEH *Lepr[Cre]*-only Control animals.

The online version of this article includes the following source data for figure 4:

**Source data 1.** Raw data from chronic CNO dosing.

expenditure on high-fat chow and become more obese than controls (***Bingham et al., 2008***; ***Dhillon et al., 2006***; ***Gonçalves et al., 2014***; ***Senn et al., 2019***). These studies thus suggest a role for leptin action on LepRb[VMH] cells in the control of energy expenditure, but were unable to study LepRb[VMH] cells more broadly. In contrast, our use of the tTARGIT system, together with *Slc17a6[FlpO]*; *Lepr[Cre]* mice, identified the projections of LepRb[VMH] cells, demonstrated their ability to acutely suppress food intake as well as promoting energy expenditure (identifying BAT thermogenesis as a target for these cells), and revealed the diurnal nature of LepRb[VMH] neuron-mediated control of energy balance. Presumably, the finding that LepRb[VMH] neuron activation alters food intake and energy expenditure specifically during the light cycle suggests that these neurons may decrease in activity during the inactive/light phase, permitting us to observe the effects of artificial neuron activation during this time.

In summary, we have developed a suite of dyad AAV vectors for the study of intersectional neural populations marked by the co-expression of Flp and Cre or marked by expression of Flp in the absence of Cre. This tTARGIT system yields robust dual recombinase-sensitive expression of the desired payload in vivo. With this approach, we defined the neural circuitry and functional capacity of LepRb$^{VMH}$ neurons. These intersectional genetic tools will facilitate the study of a broad range of dual gene-defined cell populations across the central nervous system.

# Materials and methods

## Key resources table

| Reagent type (species) or resource | Designation | Source or reference | Identifiers | Additional information |
|---|---|---|---|---|
| Antibody | Anti-FOS (Rabbit monoclonal) | Cell Signaling | Cat# 2250; RRID:AB_2247211 | (1:1000) |
| Antibody | Anti-GFP (Chicken polyclonal) | Aves Laboratories | Cat# GFP1020; RRID:AB_10000240 | (1:1000) |
| Antibody | Anti-dsRed (Rabbit polyclonal) | Takara | Cat # 632496; RRID:AB_10013483 | (1:1000) |
| Cell line (E. coli) | Stbl3 E. coli | Thermofisher | Cat # C737303 | |
| Recombinant DNA reagent | pAAV-nEF Con/Fon hChR2(H134R)-EYFP | Fenno et al., 2014; Addgene | Addgene Plasmid ID:55644 | |
| Recombinant DNA reagent | AAV-hSYN1-fDIO-tTA | This manuscript | Addgene plasmid ID: 166597 | hSYN1-driven, Flp-dependent expression of tTA |
| Recombinant DNA reagent | AAV-TRE-DIO-hM3Dq-mCherry (Cre-ON) | This manuscript | Addgene plasmid ID: 166599 | TRE-driven, Cre-dependent expression of hM3Dq-mCherry |
| Recombinant DNA reagent | AAV-TRE-DIO-hM3Dq-mCherry (Cre-OFF) | This manuscript | Addgene plasmid ID: 166609 | TRE-driven, Cre-dependent inactivation of hM3Dq-mCherry |
| Recombinant DNA reagent | AAV-TRE-DIO-ChR2-Tdtomato | This manuscript | Addgene plasmid ID: 166600 | Cre-dependent expression of ChR2-TdTomato downstream of a TRE |
| Recombinant DNA reagent | AAV-TRE-DIO-GFP-2A-SynmRuby | This manuscript | Addgene plasmid ID:166608 | Cre-dependent expression of GFP-2A-Synaptophysin-mRuby downstream of a TRE |
| Chemical compound, drug | CNO | Tocris | Cat# 4936 | |
| Genetic reagent (M. musculus) | Lepr$^{IRES-Cre}$ | Jackson Laboratories | Strain #: 032457 | Mouse: Lepr-IRES-Cre |
| Genetic reagent (M. musculus) | Slc17a6$^{IRES-FlpO}$ | This manuscript | N/A | Slc17a6 FlpO knockin mouse |
| Genetic reagent (M. musculus) | C57Bl6J | Jackson Laboratories | Strain #: 000664 | Wild-type animals |
| Genetic reagent (M. musculus) | Rosa26$^{LSL-eGFP-L10a}$ | Krashes et al., 2014 | N/A | Rosa26 targeted Cre-dependent eGFP_L10a allele |
| Genetic reagent (M. musculus) | Rosa26$^{FSF-eGFP-L10a}$ | This manuscript | N/A | Rosa26 targeted Flp and flp-dependent eGFP_L10a allele |
| Genetic reagent (M. musculus) | Rosa26$^{RCFL-eGFP-L10a}$ | This manuscript | N/A | Rosa26 targeted Flp-dependent eGFP_L10a allele |
| Software, algorithm | Prism V8 | Graphpad | https://www.graphpad.com/scientific-software/prism/ | |
| Software, algorithm | ImageJ | NIH | https://imagej.nih.gov/ij/download.html | |

### Lead contact

Further information and requests for resources and reagents should be directed to and will be fulfilled by the Lead Contact, Martin Myers Jr (mgmyers@med.umich.edu).

### Material availability

Plasmids generated by this study have been deposited to Addgene.

### Animals

Mice were bred in the Unit for Laboratory Animal Medicine at the University of Michigan. These mice and the procedures performed were approved by the University of Michigan Committee on the Use and Care of Animals and in accordance with Association for the Assessment and Approval of Laboratory Animal Care and National Institutes of Health guidelines. Unless otherwise indicated, mice were provided with ad libitum access to food (Purina Lab Diet 5001) and water in temperature-controlled (25°C) rooms on a 12 hr light–dark cycle with daily health status checks. *Rosa26* $^{LSL-eGFP-L10a}$ mice (*Krashes et al., 2014*) and *Lepr*$^{Cre}$ mice (*Leshan et al., 2006*) have been described previously. Both male and female mice were used for all studies. Sample size was determined based on previous experience.

### Generation of *Slc17a6*$^{FlpO}$ mouse line

*Slc17a6*$^{FlpO}$ mice were generated using recombineering techniques as previously described (*Balthasar et al., 2005*). Briefly, the FlpO transgene (Addgene plasmid #13793) and a LoxP-flanked neomycin selection cassette were subcloned after an optimized internal ribosome entry sequence (IRES). The IRES-FlpO-neomycin cassette was then targeted 3 bp downstream of the stop codon of *Slc17a6* in a bacterial artificial chromosome. The final targeting construct containing the *Slc17a6*-IRES-Flpo neomycin cassette and 4 kb of flanking genomic sequence on both sides was electroporated into ES cells followed by neomycin selection. Appropriately targeted clones were identified by quantitative PCR and confirmed by southern blot analysis. Targeted clones were expanded and injected into blastocysts by the University of Michigan Transgenic Core. Chimeric offspring were then bred to confirm germline transmission of the *Slc17a6*-IRES-Flpo allele; the neomycin selection cassette was removed by breeding to the E2A-Cre deleter strain (Jax stock #003724).

### Generation of Cre+Flp-dependent *Rosa26*$^{RCFL-eGFP-L10a}$ and Flp-dependent *Rosa26*$^{RFL-eGFP-L10a}$ mice

The targeting vector was developed by the Allen Brain Institute and obtained from AddGene (plasmid #61577). The neomycin resistance cassette and tdTomato sequence were removed and replaced with the eGFP:L10a coding sequences. The plasmid was then microinjected by the University of Michigan Transgenic Core into fertilized oocytes with Cas9 protein and gRNAs targeting the *Rosa26* locus (actccagtctttctagaaga). Tail DNA from the resulting pups was screened with PCR for the presence and proper insertion of the targeting vector. The Flp-dependent *Rosa26*$^{RFL-GFP-L10a}$ mouse was generated by germline deletion of the lox-stop-lox cassette.

### Stereotaxic surgery

Mice were anesthetized with isoflurane (2%) and mounted in a stereotaxic frame (Kopf). Using standard surgical techniques, 150 nL of virus was injected bilaterally via a glass micropipette attached to a microinjector (picospritzer II) targeting the VMH (Anterior/Posterior −1.3 mm; Medial/Lateral ±0.25 mm, Dorsal/Ventral −5.55 mm, relative to bregma).

For DREADD studies, hit sites were verified by mCherry detection (DSRed-IR) following euthanasia. Any data from mice in which mCherry was not detected within the VMH or was detected in other hypothalamic nuclei were discarded. Data from mice with either unilateral or bilateral viral hits were included.

### tTARGIT AAV plasmid generation

To generate the AAV-hSyn1-TetOx2-fDIO-tTA, the tTA transgene was placed within a fDIO cassette (Addgene plasmid #55641, a gift from Karl Deisseroth). The tTA sequence was then removed from pAAV-ihSyn1-tTA (Addgene plasmid #99120, a gift from Viviana Gradinaru) and replaced with the

fDIO-tTA sequence. Similarly, AAV-hSyn1TetOx2-fDIO-rtTA was generated by first placing the rtTA sequence (using rtTA sequence from Addgene plasmid #102423, a gift from Kian Peng Koh) in a fDIO cassette (Addgene plasmid #55641, a gift from Karl Deisseroth). The tTA sequence was then removed from pAAV-ihSyn1-tTA (Addgene plasmid #99120, a gift from Viviana Gradinaru) and replaced with the fDIO-rtTA sequence.

To generate the payload viruses, the GFP cassette was removed from AAV-TRE-DIO-GFP (Addgene plasmid #65449, a gift from Hongkui Zeng) and replaced with ChR2-TdTomato (Addgene plasmid #18917, a gift from Scott Sternson), hM3Dq-mCherry (Addgene plasmid #44361, a gift from Bryan Roth), GFP-2A-SynmRuby (Addgene plasmid #71760, a gift from Liqun Luo), HA-Cas9 (Addgene plasmid #61592, a gift from Feng Zhang), eGFP-L10a (provided by DPO [*Allison et al., 2015*]), Caspase3-2A-TEVp (Addgene plasmid #45580, a gift from Nirao Shah), GCaMP6s (Addgene plasmid #100845, a gift from Douglas Kim), SwiChRca-TS-YFP (Addgene plasmid #55631, a gift from Karl Deisseroth), or TVA-mcherry+oG (a gift from Marco Tripodi [*Ciabatti et al., 2017*]).

## Immunostaining

For control experiments presented in *Figure 1*, mice were euthanized 3 weeks following viral delivery. Hit sites were verified using a marker virus (AAV-CMV-Cas9-HA(18)).

Upon the completion of DREADD studies, mice were injected with CNO (1 mg/kg), sacrificed 2 h post-injection, and then perfused with 10% formalin. Brains were then removed and post-fixed in 10% formalin for 24 hr, before being moved to 30% sucrose for 24 hr. Brains were then sectioned as 30 µm thick free-floating sections. Immunohistochemical and immunofluorescent staining was performed using standard procedures using anti-FOS (1:1000, #2250, Cell Signaling Technology), GFP (1:1000, #1020, Aves Laboratories), and DSRed (1:1000, #632392, Clontech) antibodies. Images were collected on an Olympus BX51 microscope.

## Indirect calorimetry studies

Mice were singly housed 1 week prior to indirect calorimetry studies. Mice were placed into metabolic (CLAMS) cages in the University of Michigan Mouse Metabolic Phenotyping Center (UM-MMPC) and further equilibrated for 24 hr. Subsequently, mice were then injected twice daily (930 AM and 5 PM) with vehicle for 2 days followed by an additional 2 days of twice daily (930 AM and 5 PM) CNO (1 mg/kg). Data is presented as the average of the two saline days compared to the average of the two CNO days.

## Effect of CNO on energy balance

Mice were singly housed for 1 week prior to study. Mice were given ad libitum access to standard drinking water for 48 hr. For the subsequent 72 hr, standard water was replaced by water containing CNO (2.5 mg/100 mL) and 1% glucose (to make the CNO palatable). Vehicle-treated mice received 1% glucose-containing drinking water (lacking CNO) on CNO treatment days. CNO-laced water was changed daily. For the final 48 hr, mice were returned to standard drinking water. Body weight, food mass, and water levels were recorded daily. Virus controls for this study were $Slc17a6^{WT};Lepr^{Cre}$ mice co-injected with AAV-hSYN1-fDIO-tTA and AAV-TRE-DIO-hM3Dq-mCherry.

### Intrascapular temperature measurements

The UM-MMPC placed temperature transponders (IPTT-300 model with corresponding DAS-7007R reader, Bio Medic Data Systems) in the intrascapular subcutaneous tissue directly under the conjunction part of the butterfly-shaped BAT under isoflurane anesthesia. Mice were allowed to recover for 14 d before testing. One day prior to testing, ambient temperatures were increased from 22°C to 30°C. On the day of testing, mice were randomized to either CNO (1 mg/kg) or saline injections, and temperatures were recorded at −10, 10, 20, 30, 45, 60, and 120 min relative to injection time. Following 1 week, the experiment was repeated and treatment conditions (vehicle or CNO) reversed.

### Optogenetic stimulation

A single fiber-optic cannula (Doric Lenses) was implanted above the VMH (A/P: 1.3 mm, M/L: 0.25 mm, D/V: 5.0 mm) and affixed to the skull using Metabond (Fisher). After 3 weeks recovery from surgery, mice were then subjected to optical stimulation using 473 nm wavelength laser using 20 mW/

mm$^2$ irradiance. Light pulses were delivered by 1 s of 20 Hz photo stimulation and 3 s resting with multiple repetitions for 1 hr.

## Viral packaging

The INTERSECT pAAV-nEF Con/Fon hChR2(H134R)-EYFP plasmid (*Fenno et al., 2014*) was procured through Addgene (plasmid #55644). All rAAV viruses were made at the University of Michigan Vector Core using ultracentrifugation through an iodixanol gradient. rAAVs were washed three times with PBS using Amicon Ultra Centrifugal Filter Units (Millipore) and resuspended in PBS + 0.001% Pluronic F68. All viruses were packaged as AAV8 serotypes. Titers were assessed by qPCR.

| Virus | Titer (vg/mL) |
| --- | --- |
| AAV8-hSYN1-fDIO-tTA | 2.60E+13 |
| AAV8-TRE-DIO-hM3Dq-mCherry | 2.05E+13 |
| AAV8-TRE-DIO-ChR2-TdTomato | 5.83E+13 |
| AAV8-TRE-DIO-GFP-2A-SynmRuby | 5.69E+13 |
| AAV8-TRE-DIO-hM3Dq-mCherry (Cre-OFF) | 6.86E+13 |
| AAV8-nEF Con/Fon hChR2(H134R)-EYFP | 4.75E+13 |

## Statistical analysis

All data is displayed as mean ± SEM. Replicate number is included in each figure legend. Statistical analysis was performed in either Graphpad Prism eight using either t-tests or ANOVAs with Dunnet's post hoc test or linear mixed model. $p < 0.05$ was considered significant.

## Acknowledgements

We would like to acknowledge technical assistance from the University of Michigan Vector Core and the UM-MMPC, as well as support from the Molecular Genetics Core of the Michigan Diabetes Research Center (P30 DK020572). We thank members of the Myers and Olson labs and colleagues at Novo Nordisk for helpful discussions. PVS was supported by the American Diabetes Association (1–19-PDF-099), and JW was supported by a fellowship from the China Scholarship Council (201908420207). This work was supported by NIDDK DK056731 (to MGM), a Novo Nordisk Postdoctoral Project (to MGM), and NIDDK DK104999 (to DPO).

## Additional information

### Competing interests

Chien Li: CL is an employee of Novo Nordisk A/S. The other authors declare that no competing interests exist.

### Funding

| Funder | Grant reference number | Author |
| --- | --- | --- |
| American Diabetes Association | 1-19-PDF-099 | Paul V Sabatini |
| China Scholarship Council | 201908420207 | Jine Wang |
| National Institute of Diabetes and Digestive and Kidney Diseases | DK104999 | David P Olson |
| National Institute of Diabetes and Digestive and Kidney Diseases | DK056731 | Martin G Myers |
| Molecular Genetics Core | P30 DK020572 | Martin G Myers |

The funders had no role in study design, data collection and interpretation, or the decision to submit the work for publication.

## Author contributions
Paul V Sabatini, Conceptualization, Data curation, Formal analysis, Validation, Methodology, Writing - original draft, Writing - review and editing; Jine Wang, Alison H Affinati, Investigation, Writing - review and editing; Alan C Rupp, Resources, Writing - original draft, Writing - review and editing; Jonathan N Flak, Investigation; Chien Li, Conceptualization, Writing - review and editing; David P Olson, Resources, Writing - review and editing; Martin G Myers, Conceptualization, Funding acquisition, Writing - original draft, Writing - review and editing

## Author ORCIDs
Paul V Sabatini  https://orcid.org/0000-0001-6613-566X
Alan C Rupp  http://orcid.org/0000-0001-5363-4494
Martin G Myers  https://orcid.org/0000-0001-9468-2046

## Decision letter and Author response
Decision letter https://doi.org/10.7554/eLife.66835.sa1
Author response https://doi.org/10.7554/eLife.66835.sa2

## Additional files

### Supplementary files
- Transparent reporting form

### Data availability
All data generated or analysed during this study are included in the manuscript and supporting files.

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
