## [Decision Letter]

**Acceptance summary:**

It is becoming increasingly apparent that expression of single genes does not characterize specific cells; thus, intersectional approaches that require the expression of 2 genes for specificity is important. One well-established approach has been used effectively, but it failed for the application described in this paper, which fostered the development of an effective alternative method.

**Decision letter after peer review:**

Thank you for submitting your article "tTARGIT AAVs: A sensitive and flexible method to manipulate intersectional neuronal populations" for consideration by *eLife*. Your article has been reviewed by three peer reviewers, including Richard D Palmiter as the Reviewing Editor and Reviewer #1, and the evaluation has been overseen by Catherine Dulac as the Senior Editor. The following individual involved in review of your submission has agreed to reveal their identity: Bradford B Lowell (Reviewer #2). The Reviewing Editor has drafted this to help you prepare a revised submission.

Essential Revisions:

1) As Figure 1C showed, there are some leakage expression in the system with Cre. Even though the expression is very weak compared to Flp/Cre double system, the authors should discuss its limitation and possible origin. Also quantifications of the neurons labelled in these cases are important to know the level of non-specific expression.

As the authors suggested that the LepRb (Cre) is expressed in the surrounding region and difficult to restrict virus injection just to VMH, this lower Cre dependent expression of tTARGIT system would label LepRb (Cre) neurons in the surrounding regions. Then, it is difficult to conclude that the behavioral phenotype in Figures 3 and 4 are purely due to the LepRb in VMH. A control experiment might be using the *Slc17a6^WT^;Lepr^Cre^* mice.

2) The statement "promoted robust colocalization of DSRed- and FOS-IR" in Figure 1—figure supplement 4 is misleading. The c-Fos is not obviously enriched in VMH region, many c-Fos neurons are not DSRed positive and many DSRed cells are not positive for c-Fos either. Quantification is needed here.

3) The statement "almost exclusively in cells that did not express the Cre-dependent GFP (Supplementary Figure 4C)." is not true. As indicated by the arrowheads, there is quite a lot overlapping. Please provide quantification.

The original reviews are included below.

Reviewer #1 (Recommendations for the authors):

There is a robust difference in Cre- and Flp-dependent reporter expression in the VMH using the system they developed (Figure 1) versus the INTRSECT approach (Figure 1—figure supplement 2) and it was sufficient to allow viral manipulations that significantly affect food intake and energy expenditure (Figure 3). Nevertheless, quantification of the extent viral expression using the new system would a useful addition.

Please add Bregma levels to all histological figures

"..*Lepr* ablation" It is not clear at first mention whether this refers to inactivation of the Lepr gene or ablation of neurons that express Lepr. The clarification comes later.

"high-fat-diet-fed animals" add hyphens.

It would be better to refer to viruses that carry Cre-dependent genes rather that "Cre-dependent viruses" because the virus itself is not Cre dependent.

Reviewer #2 (Recommendations for the authors):

1) Serotypes of AAVs are not mentioned in the Materials and methods section.

2) Fos induction in Figure 1—figure supplement 4 is difficult to evaluate because a "no stimulation" control is not shown (or commented on).

3) As there is a general sense out there in the research community that co-injected AAVs sometimes compete (i.e. interfere with each other), do the authors have any data which they can use to address this question? Specifically, in what percentage of neurons co-expressing Cre and Flp is expression of the protein-encoded tools? Some comment about this issue in the paper could be useful.

Reviewer #3 (Recommendations for the authors):

Here are some concerns that the authors need to address:

1) Obviously, as Figure 1C showed, there are some leakage expression in the system with Cre. Even though the expression is very weak compared to Flp/Cre double system, the authors should discuss its limitation and possible origin. Also quantifications of the neurons labelled in these cases are important to know the level of non-specific expression.

As the authors suggested that the LepRb (Cre) is expressed in the surrounding region and difficult to restrict virus injection just to VMH, this lower Cre dependent expression of tTARGIT system would label LepRb (Cre) neurons in the surrounding regions. Then, it is difficult to conclude that the behavioral phenotype in Figures 3 and 4 are purely due to the LepRb in VMH. One control experiment might be using the *Slc17a6^WT^;Lepr^Cre^* mice.

2) I think "promoted robust colocalization of DSRed- and FOS-IR" in Figure 1—figure supplement 4 is not true and misleading. The c-Fos is not even obviously enriched in VMH region, many c-Fos neurons are not DSRed positive and many DSRed cells are not positive for c-Fos either. A quantification is needed here.

3).Again, I think the statement "almost exclusively in cells that did not express the Cre-dependent GFP (Supplementary Figure 4C)." is not true. As indicated by the arrowheads, there is quite a lot overlapping. Please provide quantification.

4) The Cre-off system that the authors developed does not seem to be sensitive. It may not be required for this paper, but the authors should improve their system so that a regular knock in Cre or other lower Cre level is also sufficient to turn off the expression. This would be important for the usage of this system.

---

## [Author Response]

Essential Revisions:1) As Figure 1C showed, there are some leakage expression in the system with Cre. Even though the expression is very weak compared to Flp/Cre double system, the authors should discuss its limitation and possible origin. Also quantifications of the neurons labelled in these cases are important to know the level of non-specific expression.As the authors suggested that the LepRb (Cre) is expressed in the surrounding region and difficult to restrict virus injection just to VMH, this lower Cre dependent expression of tTARGIT system would label LepRb (Cre) neurons in the surrounding regions. Then, it is difficult to conclude that the behavioral phenotype in Figures 3 and 4 are purely due to the LepRb in VMH. A control experiment might be using the Slc17a6^WT^; Lepr^Cre^ mice.

We appreciate the reviewer raising this concern. We have included a discussion of this tTA-independent expression as a limitation of the system in the revised manuscript and our expectation that this leaky expression is likely due to baseline transcriptional activity of the TRE. tTA-independent transcriptional activity from the TRE has been noted previously in other systems and represents an important consideration for any new studies using the tTARGIT system. As the reviewer suggested, we thus performed additional control experiments wherein *Lepr^Cre^*-only mice (lacking Flp expression) were injected with the tTA Driver and hM3Dq Payload AAV viruses. We have quantified the number of mCherry-IR cells within the VMN and included this data as a new supplementary figure (Figure 1—figure supplement 3). Note that the level of mCherry-IR per cell, not just the number of cells, was much lower in *Lepr^cre^* than in *Slc17a6^FlpO^;Lepr^cre^* mice.

Importantly, when dosed with CNO (administered in drinking water), these Cre-only animals lose approximately 2% of their body weight, an effect that was not significantly different when compared with vehicle treatment. This data has been included in the updated Figure 4. * indicates statistically significant differences between LeprVMH-Dq CNO and VEH while # indicates statistically significant differences between VEH and CNO treated Cre-only controls.

2) The statement "promoted robust colocalization of DSRed- and FOS-IR" in Figure 1—figure supplement 4 is misleading. The c-Fos is not obviously enriched in VMH region, many c-Fos neurons are not DSRed positive and many DSRed cells are not positive for c-Fos either. Quantification is needed here.

We thank the reviewer for bringing this oversight to our attention. We have amended the supplementary figure (now Figure 1—figure supplement 4) to include a proper “laser-off” controls and quantified the number of FOS;TdT colocalization and laser off and laser on conditions. The number of TdT-IR cells co-expressing FOS-IR (~40%) is lower than expected, possibly due to insufficient ChR2 expression or a failure on our part to optimize the photostimulation protocol.

3) The statement "almost exclusively in cells that did not express the Cre-dependent GFP (Supplementary Figure 4C)." is not true. As indicated by the arrowheads, there is quite a lot overlapping. Please provide quantification.

We agree with the reviewer that quantification of this study is critical. We have quantified GFP and mCherry single- and double-positive cells across three VMN sections per mouse and included the quantification (normalized to either the total number of VMN GFP-IR cells or the total number of VMN mCherry-IR cells) in the revised supplementary figure (now Figure 1—figure supplement 5). As presented within the figure, we observed colocalization of GFP-IR and mCherry-IR in ~6% of GFP-IR VMN cells.

We believe the main reason for this is the low level of Cre expression driven from the Lepr-IRES-Cre allele, which may have been exacerbated by insufficient viral incubation time. For our Cre-OFF studies we perfused animals 21 days post-injection, in order to be consistent with the Cre-ON/Flp-ON expression studies performed in Figure 1. However, it is possible that inactivation of the Cre-OFF payload with a weak Cre allele may require longer incubation periods. We have noted this in the Discussion.

The original reviews are included below.Reviewer #1 (Recommendations for the authors):There is a robust difference in Cre- and Flp-dependent reporter expression in the VMH using the system they developed (Figure 1) versus the INTRSECT approach (Figure 1—figure supplement 2) and it was sufficient to allow viral manipulations that significantly affect food intake and energy expenditure (Figure 3). Nevertheless, quantification of the extent viral expression using the new system would a useful addition.

We have included quantification of mCherry-IR+ VMN cells below and as part of a new supplementary figure, Figure 1—figure supplement 3. Please see full discussion above.

Please add Bregma levels to all histological figures."..Lepr ablation" It is not clear at first mention whether this refers to inactivation of the Lepr gene or ablation of neurons that express Lepr. The clarification comes later."high-fat-diet-fed animals" add hyphens.It would be better to refer to viruses that carry Cre-dependent genes rather that "Cre-dependent viruses" because the virus itself is not Cre dependent.

We thank the reviewer for their suggested improvements to the text, which we have addressed in the revised manuscript.

Reviewer #2 (Recommendations for the authors):1) Serotypes of AAVs are not mentioned in the Materials and methods section.

This was an oversight on our part. We have amended the Materials and methods section.

2) Fos induction in Figure 1—figure supplement 4 is difficult to evaluate because a "no stimulation" control is not shown (or commented on).

Failure to include the light-off ChR2 controls (images and quantification of FOS) was another oversight on our part. We have amended the figure (now Figure 1—figure supplement 4). See full discussion above.

3) As there is a general sense out there in the research community that co-injected AAVs sometimes compete (i.e. interfere with each other), do the authors have any data which they can use to address this question? Specifically, in what percentage of neurons co-expressing Cre and Flp is expression of the protein-encoded tools? Some comment about this issue in the paper could be useful.

We are aware of the argument that co-injecting AAVs can lead to competition for both the surface receptors required for internalization and the cellular machinery required for transgene synthesis and included this within the Discussion.

However, as other groups have successfully transduced neurons (and other cell types) with two AAVs (Yang et al., 2016, Omar et al., 2016, Xu et al., 2018) and when quantified, two co-injected AAVs seem to infect the same cells at very similar rates (Vardy et al., 2016 and Al-Moyed et al., 2019), we conclude using dual viruses for transfection represents a tractable approach.

Reviewer #3 (Recommendations for the authors):Here are some concerns that the authors need to address:1) Obviously, as Figure 1C showed, there are some leakage expression in the system with Cre. Even though the expression is very weak compared to Flp/Cre double system, the authors should discuss its limitation and possible origin. Also quantifications of the neurons labelled in these cases are important to know the level of non-specific expression.As the authors suggested that the LepRb (Cre) is expressed in the surrounding region and difficult to restrict virus injection just to VMH, this lower Cre dependent expression of tTARGIT system would label LepRb (Cre) neurons in the surrounding regions. Then, it is difficult to conclude that the behavioral phenotype in Figures 3 and 4 are purely due to the LepRb in VMH. One control experiment might be using the Slc17a6^WT^; Lepr^Cre^ mice.

We thank the reviewer for their comment. As this comment and comments #2 and 3 were included in the essential revisions we have responded to each in full above.

2) I think "promoted robust colocalization of DSRed- and FOS-IR" in Figure 1—figure supplement 4 is not true and misleading. The c-Fos is not even obviously enriched in VMH region, many c-Fos neurons are not DSRed positive and many DSRed cells are not positive for c-Fos either. A quantification is needed here.

See full response above.

3) Again, I think the statement "almost exclusively in cells that did not express the Cre-dependent GFP (Supplementary Figure 4C)." is not true. As indicated by the arrowheads, there is quite a lot overlapping. Please provide quantification.

See full response above.

4) The Cre-off system that the authors developed does not seem to be sensitive. It may not be required for this paper, but the authors should improve their system so that a regular knock in Cre or other lower Cre level is also sufficient to turn off the expression. This would be important for the usage of this system.

We agree with the reviewer that quantification of the Cre-OFF tTARGIT approach is a particularly salient analysis. Following quantification, we found between 2-9% of GFPIR VMN neurons also expressed the Cre-OFF hM3Dq-mCherry. Similarly, when normalized to mCherry-IR cells, between 17% of mCherry-IR cells colocalized with GFP-IR.

We will continue to design other AAV approaches to target intersectional populations but believe the Cre-OFF tTARGIT AAVs will be valuable for those interested in interrogating Flp+;Cre- cells. We have done our best to present the caveats in the revised manuscript and outline our expectation that other researchers perform proper due diligence in their models.